# Investigating the anti-osteosarcoma effects of Patchouli alcohol through protein network mapping and in vitro experiments

Zeyu Zhan[1]☯, He Pang[1]☯, Hang Wu[1], Lijun Song[2]*, Bo Wei[1]*

1 Orthopedics Center, Affiliated Hospital of Guangdong Medical University, Zhanjiang, China,
2 Reproductive Medicine Center, Affiliated Hospital of Guangdong Medical University, Zhanjiang, China

☯ These authors contributed equally to this work.
* lijunsong0369@163.com (LS); webjxmc@163.com (BW)

## Abstract

Osteosarcoma is one of the most common malignant tumors in orthopedics, especially in the metaphysis of tubular bones of the extremities in adolescents. Patchouli alcohol (PA) is a tricyclic sesquiterpene isolated from Patchouli, a Labiatae family, which has been shown to have antitumor efficacy against a variety of cancers. However, the effects and mechanisms of PA against osteosarcoma remain to be elucidated. Sixty-three possible therapeutic targets of PA against osteosarcoma were identified by protein network mapping analysis. PPI network analysis and KEGG functional enrichment analysis showed that PA exerts significant therapeutic effects on osteosarcoma through multiple targets and pathways, especially the PI3K/Akt pathway. Molecular docking results indicated that PA had significant binding to EGFR, HSP90AA1, ESR1 and SRC targets. In vitro experiments, PA inhibited proliferation and induced $G_2/M$ arrest in osteosarcoma cells in a dose- and time-dependent manner. PA induced apoptosis in osteosarcoma cells through a decrease in mitochondrial membrane potential. Alterations in the expression of apoptosis-related proteins Bcl-2 and Bax also confirmed that PA promoted apoptosis in osteosarcoma cells. Treatment with PA promoted more autophagosome formation, increased autophagy-related protein LC3-II/I ratio as well as decreased p62 expression in osteosarcoma cells. In addition, PA decreased the expression of p-PI3K/PI3K, and p-Akt/Akt in osteosarcoma cells. These findings indicated that PA exerts anti-osteosarcoma effects by inhibiting osteosarcoma cell proliferation, inducing apoptosis, promoting autophagy and inhibiting PI3K/Akt pathway, suggesting PA as a potential agent against osteosarcoma.

**Data availability statement:** All relevant data are within the manuscript and its Supporting information files.

**Funding:** This work was supported by the Clinical Study on the Elasticity Change of Paravertebral Muscles After Different Lumbar Spine Operations and Postoperative Low Back Pain (2020B01275, Dr Bo Wei); the Exploration of the Mechanism of Action of C5 Nerve Root Palsy Occurring After Anterior Cervical Approach Based on 3D Printing Technology (2021A05234, Dr Bo Wei); and the Pathogenesis and Clinical Study of Sacroiliac Joint-derived Lower Back Pain After Lumbar Spine Surgery Based on 3D Printing Technology (2022A01176, Dr Bo Wei).

**Competing interests:** The authors have declared that no competing interests exist.

## Introduction

Osteosarcoma (OS) is the most common malignant bone cancer, the most common occurrence in children and adolescents, and has a great metastatic capacity [1]. The most commonly used clinical treatment regimen is the combination of surgery with neoadjuvant chemotherapy [2], but high recurrence rates and survival rates remain a major problem [3], so that new therapeutic strategies to improve the treatment of osteosarcoma are urgently needed. Many Chinese herbal medicines have been shown to have anticancer activity, and their extraction of natural small-molecule compounds has been proved to have great potential for drug discovery [4]. Recent studies show that natural small molecule compounds derived from Chinese medicines have the potential as new antitumor agents [5]. Patchouli alcohol (PA) (C15H26O) is a tricyclic sesquiterpene, which is one of the most important bioactive ingredients in patchouli oil, and it is an important chemical marker compound in the quality control of patchouli medicine [6]. Although Patchouli alcohol has been proved to have multiple beneficial pharmacological effects, such as immunomodulatory [7], anti-inflammatory [8], antioxidant [9], anti-tumor [10], antibacterial [11], insecticidal [12], atherosclerotic [13], anti-emetic [14], whitening [15] and sedation [16]. PA follows a two-compartment model with first-order elimination, exhibiting a short half-life after intravenous and oral administration in rats. It undergoes metabolism in the liver into hydroxylated metabolites. Safety studies classify PA as low-toxicity, with an oral maximum tolerated dose (MTD) of 12.5 g/kg in mice, and LD50 values of 3.145 g/kg (intraperitoneal) and 4.693 g/kg (intragastric), suggesting its potential for therapeutic applications [15].

Apoptosis is a series of cell death processes where cells are stimulated by the specified intracellular or extracellular signals under gene regulation. Apoptosis is mainly characterized by nuclear-cytoplasmic condensation [17], nucleolar fragmentation [18], DNA degradation [19], and the formation of the apoptosome [20]. Another particular type of cell death mechanism is called autophagy. Dynamic autophagy process is strictly regulated by some specific mechanisms [21], and this catabolic process is mainly characterized by the degradation of cytoplasmic components [22], and is regulated by certain sequential steps: 1) Cell stress-induced initiation of autophagic progression [23]; 2) Phagophore form around damaged organelles; 3) Forming autophagosomes by elongating and closing phagophore; 4) Degrading damaged organelles by integrating autophagosomes and lysosomes to form autolysosome [24]. According to many studies, autophagy plays a dual role in promoting cell survival and chemosensitivity [25]. On the one hand, tumor cells recover or even lyse damaged or excess cytosolic content in a lysosome-dependent manner through the autophagy pathway, forming a healthy environment that promotes tumor growth [26]. On the other hand, excessive autophagy induces apoptosis [27].

In this study, Protein-Protein Interaction (PPI) Network of PA against osteosarcoma was constructed by protein network mapping, and its biological functions and signaling pathways were predicted and mined by GO and KEGG enrichment analysis. Finally, the efficacy and key signaling pathways of PA against osteosarcoma were verified by in vitro experiments, which provided new reference information for the development and application of PA against osteosarcoma.

## Materials and methods

### Analysis of osteosarcoma-target-PA network

The PubChem, PharmMapper, Swiss Target Prediction, Uniprot, OMIM, TTD, Genecards, and STRING databases are publicly available and allows researchers to download and analyze. Thus, our Ethics Committee of Affiliated Hospital of Guangdong Medical University waives ethical approval for open public databases that we used in this work. PA-structures from PubChem database were imported into PharmMapper, Swiss target prediction database and Uniprot database to predict potential PA related targets. The final PA target was obtained after removing duplicates. Targets associated with osteosarcoma were screened in the OMIM, TTD, and Genecards databases. Final osteosarcoma-related targets were obtained after removing duplicated targets. Venn diagrams of common drug-disease targets were drawn using the Venny software tool based on the database predicted PA-osteosarcoma targets. Subsequently, the osteosarcoma-target-PA interaction network was mapped using Cytoscape software.

### Protein-protein interaction (PPI) network construction

Targets of PA against osteosarcoma described above were imported into the STRING database to retrieve PPI networks. Then, the interaction network was imported into Cytoscape software to draw the PPI network graph. Core targets were screened by topological analysis based on PPI network information. Selected core targets are plotted by R4.0.5 software sorted by degree value.

### GO and KEGG enrichment analysis

The Gene Ontology (GO) and the Kyoto Gene and Genome Encyclopedia (KEGG) functional enrichment analysis of key target genes were performed by using the Bioconductor bioinformatics package based on the R software. GO and KEGG enrichment analysis results are output as bars and bubble plots.

### Molecular docking

PA structures obtained from PubChem database were imported into Chem3D software for structure optimization (hydrogenation and energy minimization). Subsequently, the optimized structures were imported into Schrodinger software as ligand molecules for molecular docking by establishing a database. Protein structures of EGFR (1M17), SRC (1YOL), CASP3 (2CNK), HSP90AA1 (4BQG), and ESR1 (7UJF) were obtained from the RCSB database (https://www.rcsb.org/). These proteins are optimized in the Protein Preparation Wizard module of Schrodinger software by removing crystal water, replenishing missing hydrogen atoms, repairing missing p eptides, performing energy minimization, and geometry optimization. The processing and optimization of the molecular docking are done on the Glide platform in the software. Molecular docking sites were determined based on the protein structure and ligands. Molecular pair junction bin was set to 10 Å x 10 Å x 10 Å. Finally, the molecular docking and screening were performed using the SP method [28].

### Reagents and cell culture

Human osteosarcoma cell lines HOS (cat. no. TCHu167) was purchased from the Cell Bank of the Chinese Academy of Sciences. Human osteosarcoma cell lines 143B (cat. no. CRL-8303) was obtained from American Type Culture Collection. Human osteosarcoma cell lines were cultured in MEM medium (Gibco; Thermo Fisher Scientific, Inc.) containing 10% fetal bovine serum (Gibco; Thermo Fisher Scientific, Inc.) and 1% penicillin-streptomycin (Beijing Solarbio Science & Technology Co., Ltd.)) in a cell incubator at 37°C,5% $CO_2$ and 95% humidity. The 143B and HOS cells were tested for mycoplasma contamination using the Myco-Lumi™ Luminescent Mycoplasma Detection Kit (Beyotime Institute of Biotechnology; C0298S), and all results were negative. Patchouli alcohol (purity>98%) (Shanghai Yuanye Biotechnology,

Shang-hai, China) was dissolved in DMSO. DMSO (<0.5%) was added to the control group as a vehicle control. Antibodies against Bcl-2 (cat. no. #4223), Bax (cat. no. #2772), p-ERK1/2 (cat. no. #4370), ERK1/2 (cat. no. #4695) p-PI3K (cat. no. #17366) and the secondary antibodies goat anti-rabbit IgG (cat. no. #7074) and anti-mouse IgG (cat. no. #7076) were purchased from Cell Signaling Technology. p-Akt (cat. no. ab192623), Akt (cat. no. ab179463), PI3K (cat. no. ab191606), LC3 (cat. no. ab48394), and p62 (cat. no. ab56416) antibodies were obtained from Abcam Biotechnology. The GAPDH antibody (cat. no. 21612) was purchased from Signalway Antibody.

## CCK8 assays

The concentrations of PA (25, 50, and 100 μM) were chosen according to previous studies indicating that this range exerts biological activity without detectable cytotoxicity, whereas concentrations above 100 μM tend to induce toxicity [29,30]. Osteosarcoma cells (HOS and 143B) were seeded at a density of $3.0 \times 10^3$/well in 96-well plates and grown in a thermostatic incubator for 12 h to complete attachment. Cells were then treated with various concentrations of PA (0, 25, 50, and 100 μM) for 24 and 48 hours, respectively. Each well was replaced with fresh medium containing 10% CCK8 reagent (Beyotime Institute of Biotechnology) and incubated for 2 h in a thermostatic incubator. OD values of stained cells were measured at 450 nm using a microplate reader (MK3; Thermo Fisher Scientific).

## Colony-formation assays

The 143B and HOS osteosarcoma cells were trypsinized, centrifuged, and re-suspended in fresh media, then inoculated into 6-well plates (500 cells per well) with a concentration gradient of PA. Next, the cells were cultured in the incubator for approximately 2 weeks until macroscopically visible cell clusters appeared in the dishes. The cells were fixed with 4% paraformaldehyde for 15 min and stained them with crystal violet staining solution for 15 min. We counted the number of cell clusters under a microscope (Olympus Corporation) to calculate the colony formation rates.

## Apoptosis analysis

Annexin V-FITC/PI apoptosis detection reagent (Beyotime Institute of Biotechnology) was used to assess the ratio of apoptosis in osteosarcoma cells. HOS and 143B cells treated with various concentrations of PA were resuspended with 5 μL Annexin V-FITC and 10 μL PI staining solution (Beyotime Institute of Biotechnology) and then incubated in flow tubes at room temperature for 30 min. Finally, we analyzed stained cells by flow cytometry (BD Biosciences, Franklin Lakes, NJ, USA).

## Cell cycle analysis

HOS and 143B cells were collected in flow tubes by centrifugation and fixed overnight by adding 70% chilled ethanol. Subsequently, fixed cells were stained with propidium iodide (PI) (Beyotime Institute of Biotechnology) for 30 min at room temperature. Finally, cell cycle distribution was analyzed by flow cytometry (BD Biosciences, Franklin Lakes, NJ, USA).

## Measurement of mitochondrial membrane potential (MMP)

The JC-1 kit is a fluorescent probe used to detect the mitochondrial membrane potential. When JC-1 accumulates in a matrix with high mitochondrial membrane potential, it forms polymer and generates red fluorescence. Otherwise, when membrane potential is low, JC-1 fails to accumulate in the matrix of mitochondria, when JC-1 is a monomer to generates green fluorescence.143B and HOS were collected by centrifugation and resuspended in 0.5 ml of fresh medium. Cell suspensions were stained for 30 min at room temperature according to the standard method for JC-1 working solution of the MMP assay kit (Beyotime Institute of Biotechnology). Intracellular mitochondrial fluorescence transition was detected by flow cytometry (BD Biosciences, Franklin Lakes, NJ, USA).

### Electron microscopy observe

Log-growth stage osteosarcoma cells were treated with 50 µM PA in an incubator for 24 h, then collected and washed and fixed with 3% glutaraldehyde for 12h, and finally visualized by TEM localization and photographed and recorded.

### Western blot analysis

143B and HOS cells were lysed on ice for 30 min by mixing with appropriate amounts of protein lysate (PMSF, RIPA = 1:100). Total protein concentration was determined by the BCA kit after 3 min of adequate sonication and 15 min of centrifugation in a centrifuge at 4 °C at 13,000 g/min. Add 5 × loading buffer into EP tubes containing protein, boil them in boiling water for 10 min, and dispense and store them in a −20°C refrigerator for use. Formulated 10% or 12% SDS - PAGE was used for protein (20 µg/lane) separation gel electrophoresis (100 V; 60 min) and proteins on the gel were transferred to 0.2 µm pore size PVDF membranes (Merck KGaA, Darmstadt, Germany) under current constant current (250 mA; 2.5 h). After blocking with 5% skimmed milk for 60 min, the PVDF membrane was added with the corresponding primary antibody (1:1000 diluted) and placed in a refrigerator at 4 ° C for 12 h. After washing the membrane with the pre-pared TBST solution, each PVDF membrane was incubated with the corresponding secondary antibody (1:3000 diluted) for 1 h at room temperature. Protein bands were visualized by a fluorescence imaging system (Tanon 5200; Tanon Science and Technology Co., Ltd.) combined with ECL chromogenic reagent (Zeta Life). Visualization results were analyzed by ImageJ software for gray values of bands, and expression levels of each protein were calculated using GAPDH as an internal reference.

### Statistical methods

Data processing analysis was performed with SPSS Statistics 25.0 (IBM) and GraphPad Prism 8.0 (GraphPad Software) software after each experiment was independently repeated three times. Experimental data were expressed as mean ± standard deviation (SDs). Unpaired Student's t-tests were used for comparisons between groups. Statistical differences between ≥3 groups were determined using one-way ANOVA followed by Tukey's post hoc tests. $*P < 0.05$ was considered statistically significant.

## Result

### Potential targets of PA against osteosarcoma

Based on PA-molecular structure, a total of 190 drug targets were predicted in PharmMapper database, Swiss target prediction database and Uniprot database. A total of 1230 disease targets were retrieved from the OMIM, TTD and Genecards databases using the keyword "osteosarcoma" (Fig 1A). The "osteosarcoma-target-PA " interaction network was constructed by plotting 63 common targets identified by venny plots and inputting them into Cytoscape software (Fig 1B). Sixty-three targets were entered into STRING database to retrieve network relationship data between targets resulting in PPI network graphs with 969 edges and 63 nodes. Different areas and different color depths of nodes represent degree values (Fig 1C). Subsequently, based on PPI data, we identified 30 core targets ranked based on degree value degree scores (Fig 1D). Among these core targets, EGFR (48-sided), CASP3 (47-sided), ESR1 (42-sided), HSP90AA1 (42-sided), SRC (42-sided) and IGF1 (40-sided) have high nodes, indicating that they are closely associated with PA anti-osteosarcoma dense mechanisms.

### GO and KEGG analysis of PA with osteosarcoma

Sixty-three common targets were subjected to GO enrichment to analyze biological processes, cellular components, and molecular function items. GO enrichment yielded 1444 biological process projects, 25 cellular component expression projects and 107 molecular function-related projects. Fig 2A shows the top 10 results for each part of GO enrichment,

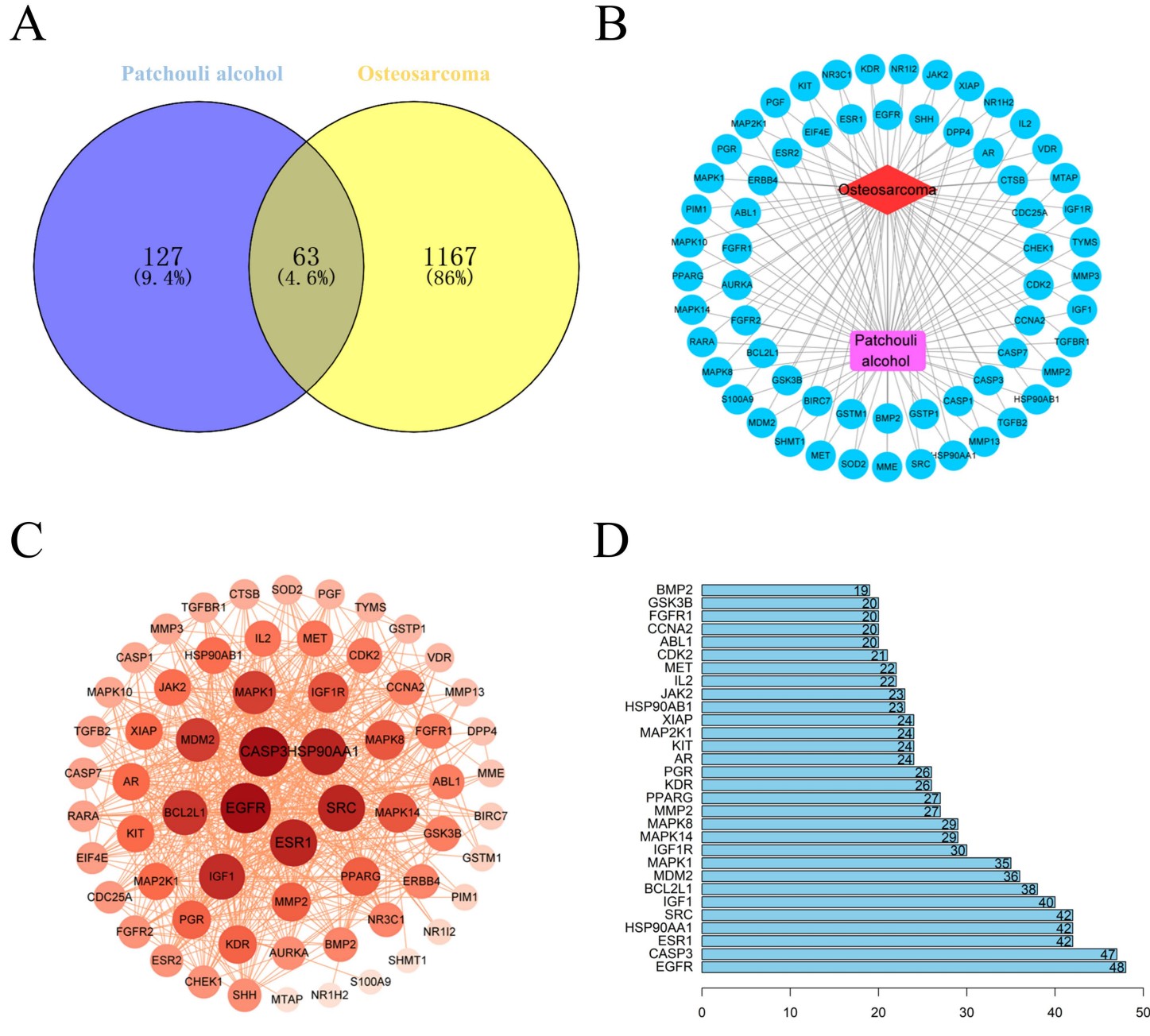

**Fig 1. Potential targets and PPI networks of PA against osteosarcoma. (A)** Venn diagram of PA and potential targets of osteosarcoma. **(B)** PA-Common target-osteosarcoma interaction network diagram. Red represents osteosarcoma. Purple represents PA and blue line represents 63 common targets. **(C)** PPI network of common targets of PA and osteosarcoma. **(D)** PPI network topology analysis showed the top 30 ranked core targets.

presented as a bar graph. P-value represents the statistical significance of GO enrichment, and the intensity of red increases as the significance level increases. Subsequently, KEGG enrichment of common targets mined 133 signaling pathways, and the pathway results of the top 20 screened were output as bar graphs (Fig 2B). KEGG enrichment results mainly included cancer-related pathways, such as PI3K/Akt pathway, MAPK pathway, Proteoglycans in cancer pathways, and Ras pathways.

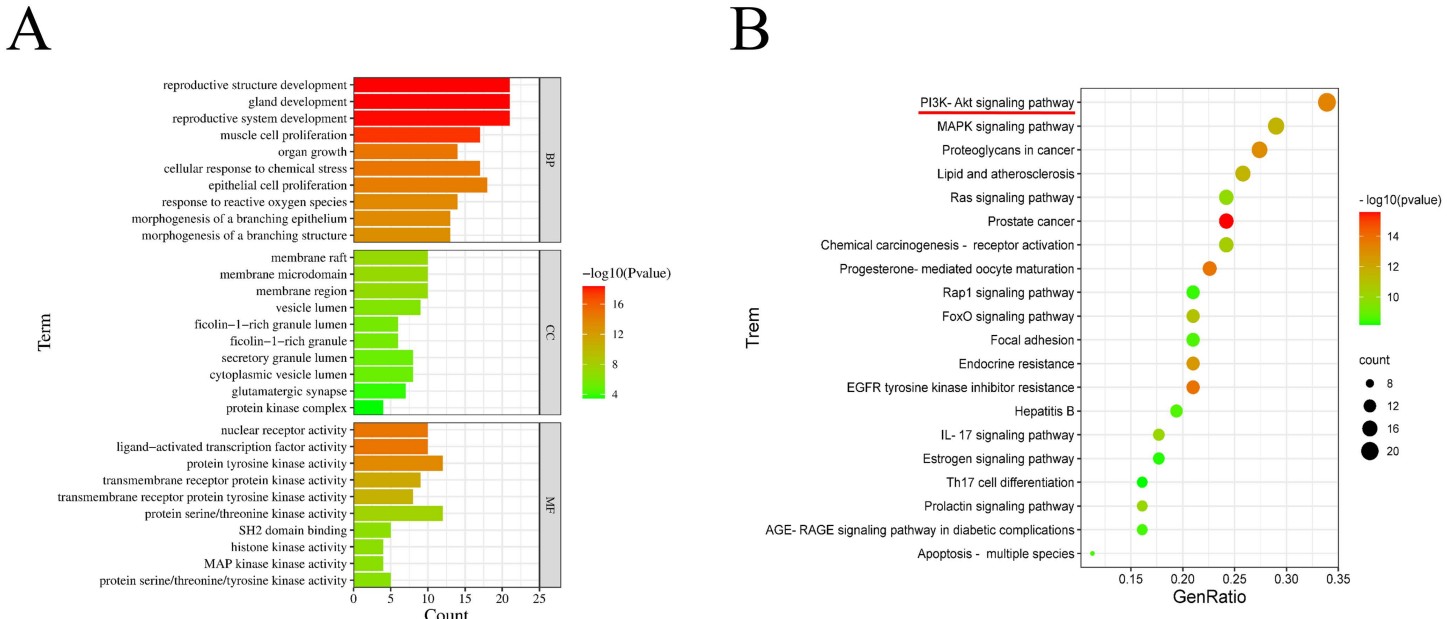

**Fig 2. GO and KEGG analysis results of PA against osteosarcoma. (A)** GO enrichment analysis of common targets of PA against osteosarcoma. **(B)** KEGG enrichment analysis of common targets. Bubble plots show the top 20 enriched pathways. The intensity of red color increases with the level of significance of P value.

## Molecular docking analysis of core targets

According to the expected core targets of the PPI network that were significantly associated with anti-osteosarcoma effects, the top five target proteins (EGFR, Caspase-3, ESR1, HSP90AA1, and SRC) ranked by degree value were subjected to molecular docking. Binding energy and hydrogen bonding interactions of molecular docking were used to evaluate the affinity of PA (Fig 3A) to target proteins. Visualization showed amino acid residues and hydrogen bonding sites in the active pocket of the PA-binding core protein. PA binds EGFR with an energy of −5.79 kcal/mol and forms hydrogen bonds with CYS-773, ASP831, and ARG-817 amino acids in the active pocket (Fig 3B). PA was embedded in the caspase-3 active pocket position with binding energy of −6.18 kcal/mol(Fig 3C). PA binds to ESR1 protein with a binding energy of −7.22 kcal/mol and forms a hydrogen bond with amino acid MET-421 in the active pocket (Fig 3D). PA binds to HSP90AA1 with an energy of −6.43 kcal/mol and forms a hydrogen bond with GLY-135 amino acid in the active pocket (Fig 3E). PA binds to SRC with an energy of −5.95 kcal/mol and forms a hydrogen bond with amino acid SER-347, ASP-350, and LEU-275 in the active pocket (Fig 3F). In conclusion, our results indicate that PA has a strong affinity and binding potential with the core targets EGFR, CASP3, ESR1, HSP90AA1, SRC proteins.

## PA inhibits proliferation and promotes G$_2$/M arrest in osteosarcoma cells

On the basis of previous experiments, the proliferation ability of HOS and 143B cells treated with PA at concentrations (0 µM, 25 µM, 50 µM, and 100 µM) for 24 h and 48 h was determined by CCK-8 assay. The results showed that PA significantly inhibited the proliferation of osteosarcoma cells in a dose- and time-dependent manner (Fig 4A). The colony formation assays of two osteosarcoma cells treated with different PA concentrations showed that PA significantly decreased the number of osteosarcoma cell colonies of the cells (Fig 4B). At the same time, the results of cell cycle arrest showed that compared with the control group, G$_2$/M phase increased and G0/G1 cells decreased in the PA group (Fig 4C). Our results revealed that PA had an anti-proliferative effect on osteosarcoma cells, which may be related to the mechanism of G$_2$/M cycle arrest.

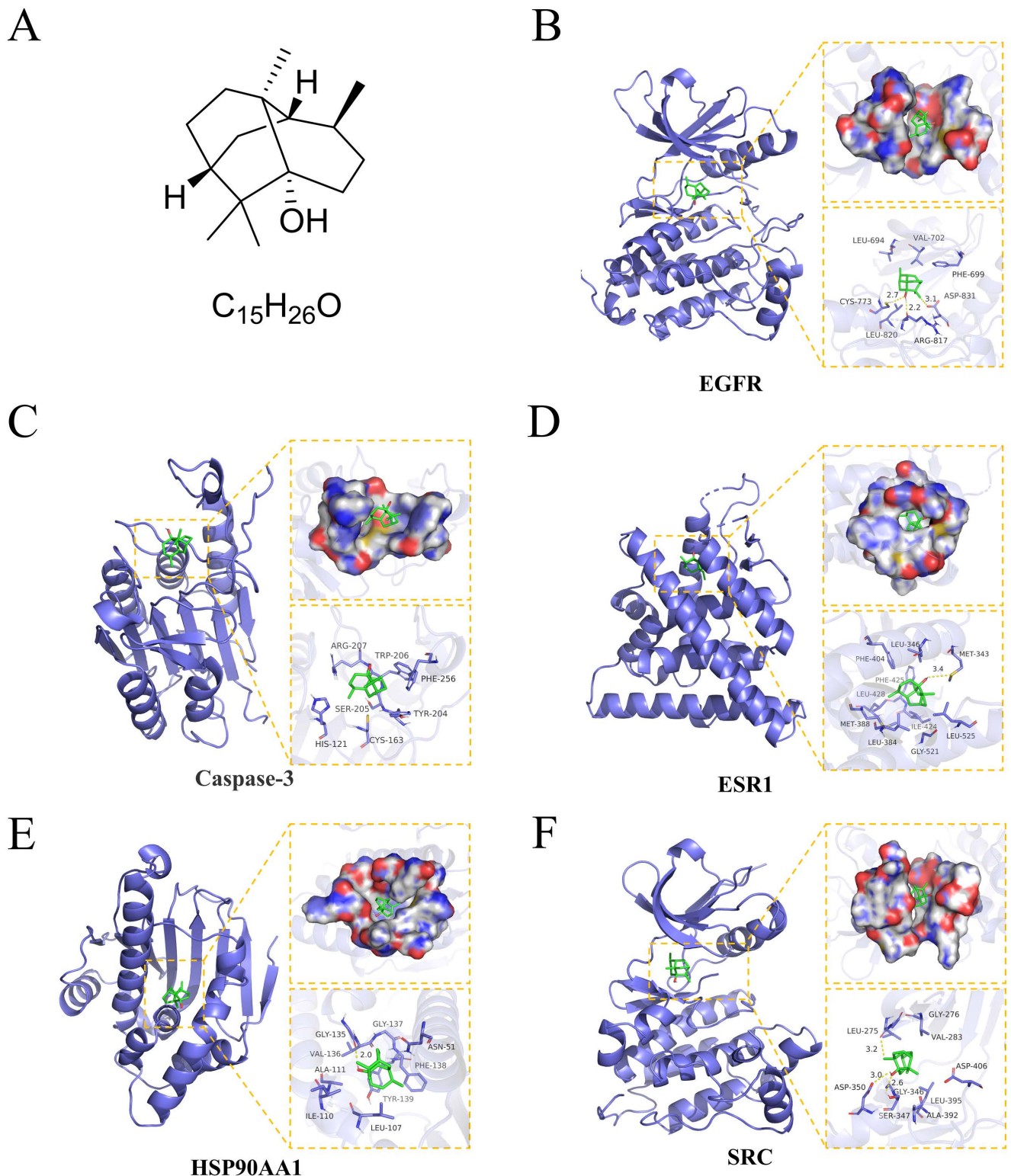

**Fig 3. Molecular Docking Analysis of PA with its core target. (A)** Chemical structure and molecular formula of PA. **(B)** PA binds three amino acid residues (CYS-773, ARG-817, and ASP-831) in the EGFR protein via hydrogen bonds. **(C)** PA binds to CASP3 protein. **(D)** PA binds to one amino acid residues (MET-343) in the ESR1 protein via hydrogen bonds. **(E)** Asiatic acid binds one amino acid residues (GLY-135) in the HSP90AA1 protein via hydrogen bonds. **(F)** PA binds three amino acid residues (LEU-275, ASP350 and GLY-346) in the SRC protein via hydrogen bonds. Yellow dashed lines represent hydrogen bonds.

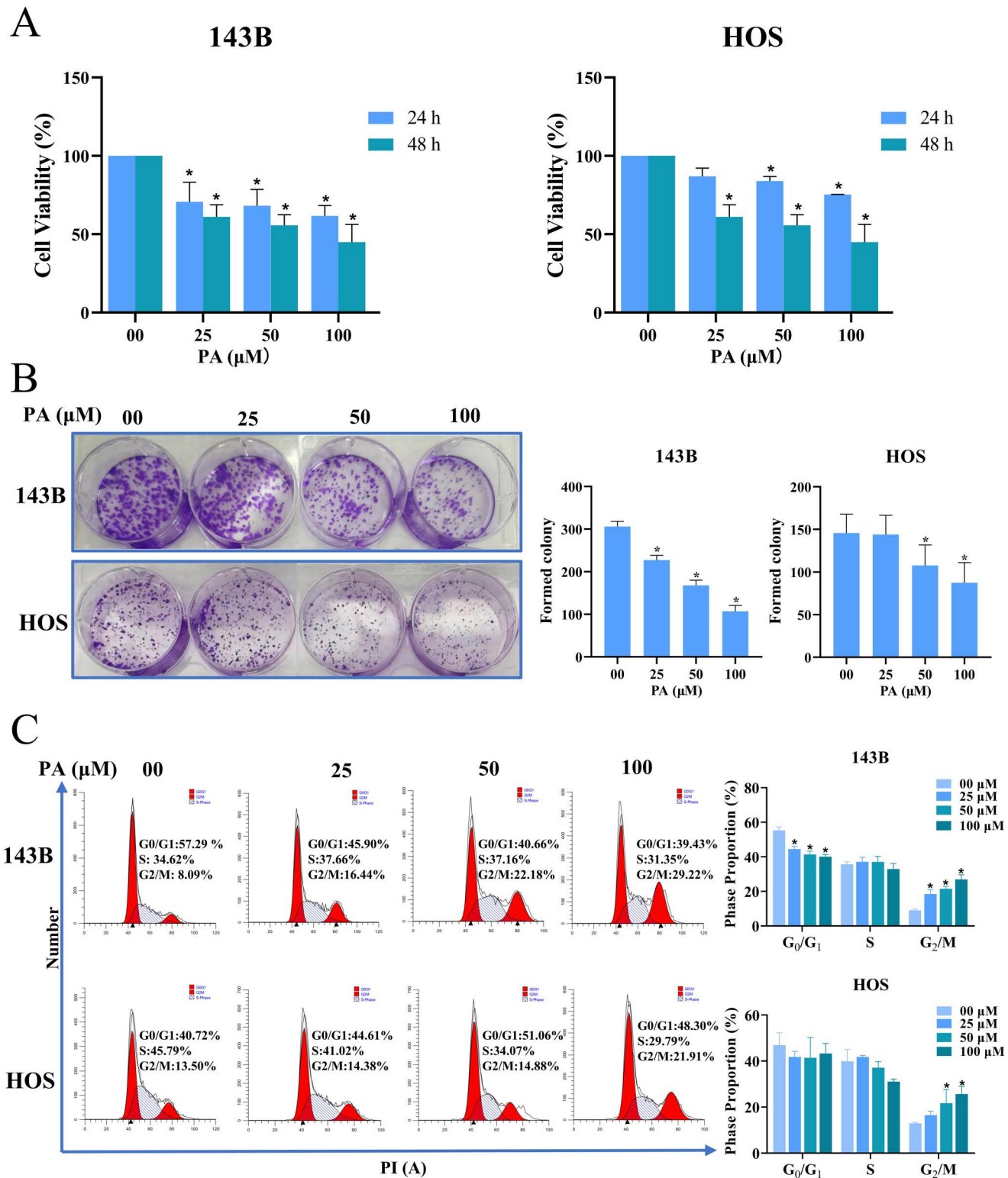

**Fig 4. PA inhibits the proliferation of osteosarcoma cells. (A)** Following treatment of osteosarcoma 143B and HOS cells with PA (00, 25, 50, and 100 μM) for 24−48 h, cell viability was determined by using CCK-8 assay. **(B)** PA inhibited colony formation ability of 143B and HOS cells. **(C)** Cycle arrest of 143B and HOS cells detected by flow cytometry. *$p < 0.05$ *vs.* control group.

## PA induced apoptosis in osteosarcoma cells

To assess the effect of PA on apoptosis in osteosarcoma cells, we detected the proportion of apoptotic cells by Annexin V-FITC/PI double fluorescence staining. In the current study, osteosarcoma cells were cultured for 24 h in different concentrations of PA, and the proportion of apoptotic cells was increased in the treated group compared with control group (Fig 5A). Decreased intracellular mitochondrial membrane potential is a marker of early apoptosis. To determine whether there was a decrease in intracellular mitochondrial membrane potential, the transition from red fluorescence to green fluorescence was detected by JC-1 staining reagent. As Fig 5B shown, with the increase of PA dose, the mitochondrial membrane fluorescence of osteosarcoma cells 143B and HOS gradually changed from red to green fluorescence, suggesting that PA induced early apoptosis of osteosarcoma cells. At the same time, the effect of PA on apoptosis of osteosarcoma cells was further verified by detecting the changes of mitochondrial apoptosis-related proteins. Western blot analysis showed that PA increased the expression of Bax protein and decrease the expression of Bcl-2 protein (Fig 5C). In summary, PA promoted mitochondrial dysfunction in osteosarcoma cells, thereby inducing apoptosis.

## PA triggers autophagy in osteosarcoma cells

To demonstrate the relationship between PA and autophagy, the expression of autophagy-related proteins LC3 and p62 was measured in HOS and 143B cells treated with different concentrations (00 µM, 25 µM, 50 µM and 100 µM) of PA for 24 h. Western blot analysis revealed that the autophagy-related protein LC3-II/I ratio was significantly increased and p62 expression was significantly decreased in osteosarcoma cells with increasing PA concentration (Fig 6A). Subsequently, an increase in the number of autophagic vesicles in osteosarcoma cells in the PA-treated group was observed by transmission electron microscopy (Fig 6B). Thus, PA triggered autophagy in osteosarcoma cells.

## PA inhibits PI3K/Akt pathway in osteosarcoma cells

Combined with the prediction of KEGG enrichment analysis, the PI3K/Akt pathway is a potential pathway of PA against osteosarcoma. As shown in Fig 7, PA significantly decreased the expression of p-PI3K/PI3K and p-Akt/Akt in osteosarcoma cells. The above results indicate that PA exerts an anti-osteosarcoma effect by inhibiting the PI3K/Akt signaling pathway.

## Discussion

At present, the treatment of osteosarcoma is still a comprehensive treatment of surgical treatment, combined with preoperative and postoperative chemotherapy, immunotherapy and others. With the development of neoadjuvant chemotherapy, the success rate of limb salvage therapy has been significantly improved [31], but 30% to 50% of local osteosarcoma will eventually recur [32]; at the same time, high-dose chemotherapeutic drugs such as methotrexate [33], doxorubicin [34], and ifosfamide [35] can bring great side effects on physical health [36]. PA exerts its anti-cancer effects through multiple pathways, including inducing tumor cell apoptosis, inhibiting invasion, arresting cell cycle progression, and modulating key signaling pathways such as Akt/mTOR and NF-κB [37,38]. Notably, combination therapy with PA and cisplatin demonstrated remarkable synergistic effects in both melanoma and lung cancer preclinical models, synergistically enhanced anti-tumor efficacy while ameliorating cisplatin-associated toxicity [39,40]. Therefore, the search for a new, effective, and safe treatment is a key issue that needs to be solved urgently. Our results reveal that 63 key proteins associated with multiple biological processes and signaling pathways associated with osteosarcoma were screened, but the functional changes of these molecules and the profound impact of therapies still need further animal experiments and clinical trials to elucidate. PPI results showed that EGFR, CASP3, ESR1, HSP90AA1, and SRC may be potential targets for PA against osteosarcoma. Molecular docking results showed that PA had a good match and affinity to them, and the binding energy was less than −5 kcal/mol [41], indicating that PA exerts anti-osteosarcoma effects through these targets.

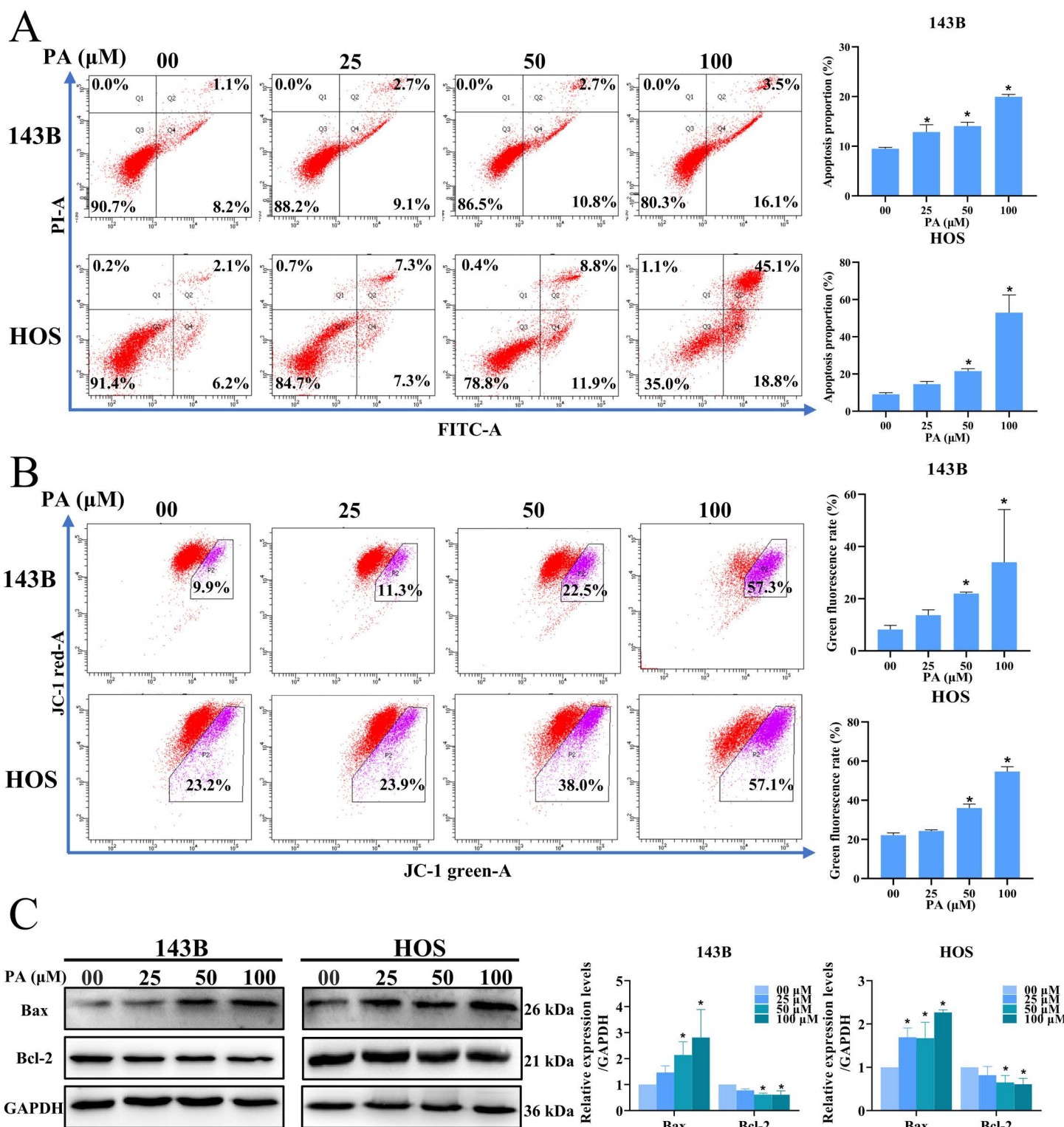

**Fig 5. PA-induced apoptosis in osteosarcoma cells. (A)**The ratio of apoptosis in osteosarcoma cells was detected by Annexin V-FITC double staining combined with flow cytometry analysis. **(B)** Intracellular MPP turnover was detected by staining with probe JC-1 combined with flow cytometry analysis. **(C)** The expression levels of apoptosis-related proteins Bcl-2 and Bax in osteosarcoma cells (HOS and 143B) were measured by Western blot analysis. *$p < 0.05$ *vs*. control group.

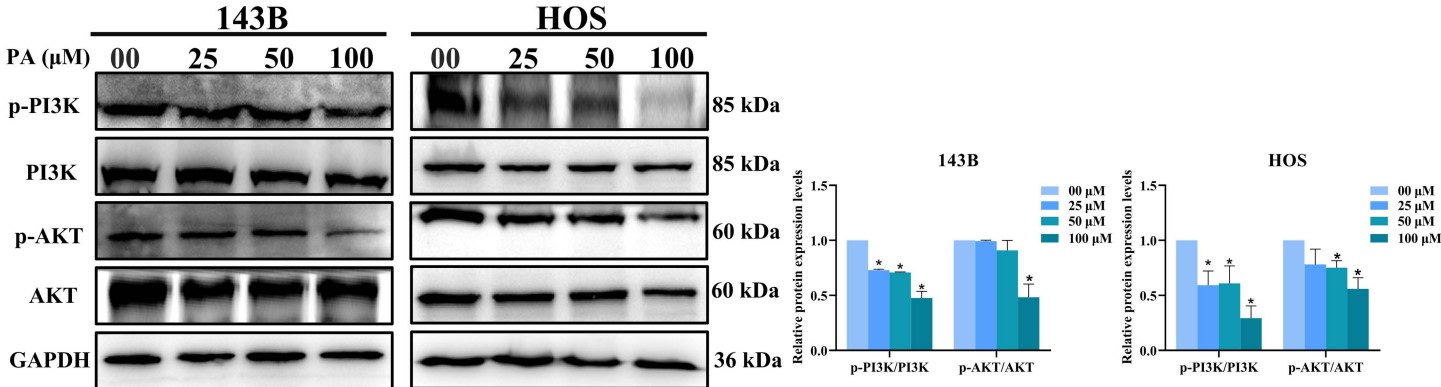

**Fig 6. PA triggers autophagy in osteosarcoma cells. (A)** PA-triggered autophagy-related protein LC3I/II and p62 protein expression was detected by Western blot analysis. **(B)** PA promoted an increased quantities of autophagosomes within 143B and HOS cells as observed by transmission electron microscopy. Red triangles point to autophagosome position. Low magnification, 8000X; High magnification, 30,000X. *$p < 0.05$ *vs.* control group.

**Fig 7. PA inhibits PI3K/AKT pathways in osteosarcoma cells.** Following treatment of 143B and HOS cells with PA (00, 25, 50, and 100 μM) for 24h, protein levels of p-PI3K, PI3K, p-Akt, and Akt as measured by western blot analysis. *$p < 0.05$ *vs.* control group.

Apoptosis is regulated by two classical apoptotic pathways, the internal pathway (mitochondria-mediated apoptosis pathway) and the extrinsic pathway (death receptor-mediated apoptosis pathway) [42]. Many studies have confirmed that the ratio of pro-apoptotic to anti-apoptotic proteins in Bcl-2 family proteins affects the sensitivity of cells to death signals [43]. Decreased mitochondrial membrane potential usually occurs in the early stages of apoptosis [44]. The imbalance of Bcl-2 family proteins leads to a decrease in mitochondrial membrane potential and the release of cytochrome C into the cytoplasm, which in turn activates the caspase family and induces mitochondria-mediated apoptosis [45]. Apoptosis is often accompanied by cell cycle arrest [46]. Unlike normal cells, cell cycle checkpoint responses are defective in tumor cells [47]. G2/M phase is the last time for cells to repair DNA damage before entering mitosis and is an important biological target for the treatment of tumors [48]. Our results indicate that PA inhibits osteosarcoma cell proliferation and induces $G_2$/M arrest. In addition, PA treatment reduced MMP and increased the proportion of apoptosis in osteosarcoma cells. Western Blot Analysis results showed that the expression of anti-apoptotic protein Bcl-2 decreased with increasing PA concentration, while the expression of pro-apoptotic protein Bax was enhanced. In summary, PA promotes mitochondrial dysfunction in osteosarcoma cells, which induces endogenous apoptosis. At the autophagosome form stage, the lipidated form LC3-II produced by the binding of phosphatidylethanolamine to cytoplasmic LC3-I is incorporated into autophagic vesicles [49]. The p62 of the linked ubiquitinated protein is selectively encapsulated into autophagosomes, which is degraded by the action of proteolytic enzymes in autolysosomes [50]. Therefore, the expression of LC3-II protein presented a positive correlation with autophagic activity [51], while p62 showed a negative correlation [52–53]. Our experimental results showed an up-regulation of the ratio of LC3-II/LC3-I and a decrease in p62 expression levels. In addition, a significant increase in autophagosomes in the PA-treated group was observed under transmission electron microscopy, confirming that PA triggered autophagy in osteosarcoma cells.

Based on the number of targets corresponding to KEGG enriched pathways, we focused on the PI3K/Akt key pathways involved in the anti-osteosarcoma mechanism of PA. Accumulating evidence suggests that the PI3K/Akt pathway is essential for cell proliferation [54], migration [55], apoptosis [56], and autophagy in osteosarcoma [57]. In addition, PI3K/Akt pathway is closely related to distant metastasis [58] and chemoresistance [59] of osteosarcoma, and inhibiting the activation of this pathway plays a role in anti-osteosarcoma. Western blot analysis results showed that the p-PI3K/PI3K and p-Akt/Akt expression levels in the PA-treated group were lower than those in the control group, indicating that PA could effectively inhibit the activation of the PI3K/Akt pathway in osteosarcoma cells. In summary, based on our findings, PA exerts anti-osteosarcoma efficacy by promoting G2/M arrest, inducing apoptosis and inhibiting the PI3K/Akt pathway.

Although PA has been studied in other cancers, its effects in OS have not previously been reported. Here, we demonstrate for the first time that PA induces both apoptosis and autophagy in OS cells while suppressing PI3K/Akt activity. By integrating network-based target prediction with experimental validation, we identified key nodes such as EGFR, HSP90AA1, and SRC, which are established contributors to OS progression and represent druggable targets. These findings expand the pharmacological profile of PA into a new disease context and provide a mechanism-based rationale for potential therapeutic combinations, including with platinum agents or pathway inhibitors. In conclusion, by uncovering multiple mechanisms, including apoptosis, autophagy, and PI3K/Akt pathway inhibition, this study expands the pharmacological profile of PA into osteosarcoma for the first time, underscoring its novelty and clinical relevance.

## Conclusion

We concluded that PA was comprehensively screened for therapeutic targets and pathways against osteosarcoma by protein network mapping and molecular docking simulations. PA mediated apoptosis and autophagy in osteosarcoma cells through inhibition of the PI3K/Akt signaling pathway as assessed by in vitro experiments. These findings reveal the potential therapeutic targets and mechanisms of PA against osteosarcoma and provide a theoretical basis for its application in osteosarcoma treatment (Fig 8). However, this study also has some limitations. Our validation process was limited to silico and in vitro experiments, without extending to in vivo studies. Furthermore, we did not include the validation of therapeutic

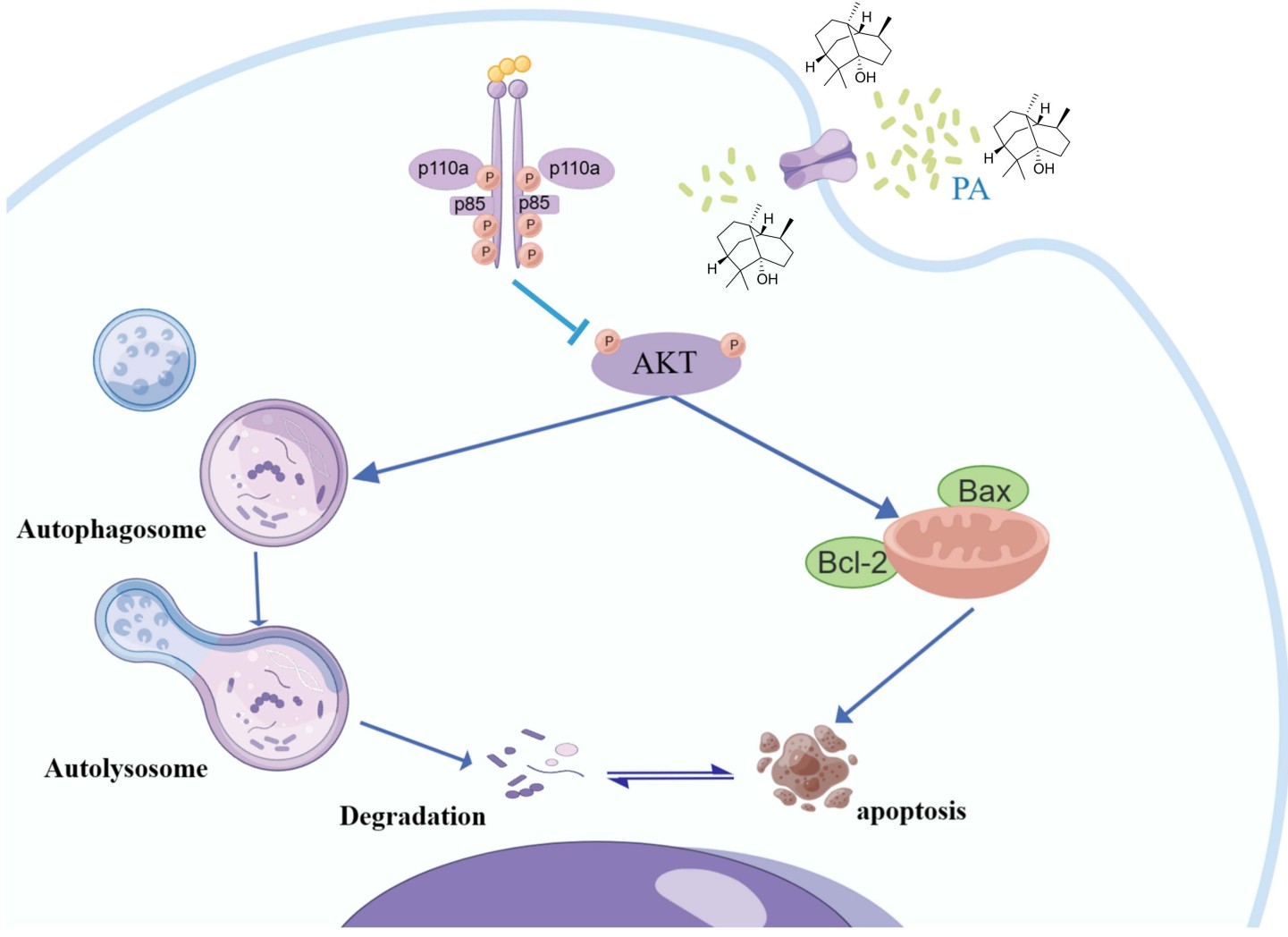

**Fig 8. Potential mechanism of anti-osteosarcoma activity of PA.**

targets, which is essential for a comprehensive assessment of efficacy. Therefore, additional in vitro and in vivo experiments are needed to further elucidate the mechanisms and effects of PA against osteosarcoma.

## Supporting information

**S1 Data. Supporting western blot data.**
(PDF)

## Author contributions

**Formal analysis:** He Pang, Zeyu Zhan.

**Funding acquisition:** Lijun Song, Bo Wei.

**Methodology:** Lijun Song, Bo Wei.

Project administration: Bo Wei.

Software: He Pang, Zeyu Zhan, Hang Wu.

Supervision: Zeyu Zhan.

Visualization: Hang Wu.

Writing – original draft: He Pang, Zeyu Zhan.

Writing – review & editing: Lijun Song, Bo Wei.

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
