## [Editor Report · Decision Letter 0]

22 Oct 2024

Dear Dr. Pang,

Thank you for submitting your manuscript to PLOS ONE. After careful consideration, we feel that it has merit but does not fully meet PLOS ONE’s publication criteria as it currently stands. Therefore, we invite you to submit a revised version of the manuscript that addresses the points raised during the review process.

Thank you for your submission. The manuscript is well-written and presents important findings. However, I would like to suggest a few revisions for improved clarity and completeness.

**Title** : Most readers may not be familiar with the term "network pharmacology." I recommend reconsidering the title to make it more accessible such as *Investigating the Anti-Osteosarcoma Effects of Patchouli Alcohol through Protein Network Mapping and In Vitro Experiments* . If you prefer to retain "network pharmacology" in the title, I suggest explaining this term early in the introduction to ensure all readers understand its relevance.**Cell Line Testing** : Please clarify whether the cell lines used in your experiments were tested for mycoplasma contamination, as this is an important aspect of ensuring the validity of in vitro studies.**Introduction** : I recommend adding a full paragraph about Patchouli Alcohol (PA) in the introduction, including details such as its half-life, route of elimination, pharmacokinetics/pharmacodynamics (PK/PD), safety, and maximum tolerated dose (MTD). This information will provide valuable context for readers.**Discussion:** Additionally, please include relevant references that discuss PA, particularly in combination with other anticancer drugs. A useful resource is PubMed, where several relevant studies can be found here. https://pubmed.ncbi.nlm.nih.gov/?term=Patchouli+alcohol+cancer&sort=date

We look forward to receiving your revised manuscript.

Kind regards,

Joseph Alan Bauer, Ph.D.

Academic Editor

PLOS ONE

Journal Requirements:

5. PLOS ONE now requires that authors provide the original uncropped and unadjusted images underlying all blot or gel results reported in a submission’s figures or Supporting Information files. This policy and the journal’s other requirements for blot/gel reporting and figure preparation are described in detail at https://journals.plos.org/plosone/s/figures#loc-blot-and-gel-reporting-requirements and https://journals.plos.org/plosone/s/figures#loc-preparing-figures-from-image-files . When you submit your revised manuscript, please ensure that your figures adhere fully to these guidelines and provide the original underlying images for all blot or gel data reported in your submission. See the following link for instructions on providing the original image data: https://journals.plos.org/plosone/s/figures#loc-original-images-for-blots-and-gels.

---

## [Author Response · Author response to Decision Letter 1]

1 Apr 2025

Dear Editors and Reviewers:

Thank you for your letter and the reviewers' insightful comments regarding our manuscript, titled "Exploration of the Anti-osteosarcoma Activity of Patchouli Alcohol Based on Network Pharmacology and In Vitro Experiments" (ID: PONE-D-24-41403). We sincerely appreciate the reviewers’ valuable feedback, which has been instrumental in improving the quality and clarity of our work. In response to the editor's and reviewers’ suggestions, we have conducted minor revisions to our manuscript and incorporated additional content to further substantiate our findings. Additionally, we have carefully revised the manuscript to ensure full compliance with the journal’s formatting guidelines. The revised sections have been highlighted in red for easy reference. We greatly appreciate your time and consideration, and we look forward to your feedback on our revised submission.

Respond to the reviewer's comments:

Question 1. Title: Most readers may not be familiar with the term "network pharmacology." I recommend reconsidering the title to make it more accessible such as Investigating the Anti-Osteosarcoma Effects of Patchouli Alcohol through Protein Network Mapping and In Vitro Experiments. If you prefer to retain "network pharmacology" in the title, I suggest explaining this term early in the introduction to ensure all readers understand its relevance.

Response: Thank you for your valuable suggestion. We understand that "network pharmacology" might be a less familiar term for some readers. We have revised the title to make it more accessible, as you suggested. The new title is: "Investigating the Anti-Osteosarcoma Effects of Patchouli Alcohol through Protein Network Mapping and In Vitro Experiments." We hope this revision addresses your concern and enhances the comprehensibility of our work. Thank you again for your thoughtful feedback!

Question 2. Cell Line Testing: Please clarify whether the cell lines used in your experiments were tested for mycoplasma contamination, as this is an important aspect of ensuring the validity of in vitro studies.

Response: Thank you for highlighting this important point. We confirm that all cell lines used in our experiments were tested for mycoplasma contamination prior to the study. These tests were conducted using Myco-Lumi™ Luminescent mycoplasma detection kit (Beyotime Institute of Biotechnology; C0298S), and all results were negative, ensuring the validity and reliability of our in vitro experiments. We have added this information to the Methods section of the manuscript for clarity and transparency.

Question 3. Introduction: I recommend adding a full paragraph about Patchouli Alcohol (PA) in the introduction, including details such as its half-life, route of elimination, pharmacokinetics/pharmacodynamics (PK/PD), safety, and maximum tolerated dose (MTD). This information will provide valuable context for readers.

Response: Thank you for your insightful suggestion. We agree that providing additional details about Patchouli Alcohol (PA) will enhance the context and understanding of our study. In response to your recommendation, we have added a dedicated paragraph in the introduction that includes information on the half-life, route of elimination, pharmacokinetics/pharmacodynamics (PK/PD), safety profile, and maximum tolerated dose (MTD) of PA. Specifically, we have incorporated relevant data from existing literature to provide a comprehensive overview of PA’s characteristics and its relevance to our investigation of its anti-osteosarcoma effects.

We believe this addition will give readers a clearer understanding of PA’s therapeutic potential and its role in our study.

Question 4. Discussion: Additionally, please include relevant references that discuss PA, particularly in combination with other anticancer drugs. A useful resource is PubMed, where several relevant studies can be found here.

https://pubmed.ncbi.nlm.nih.gov/?term=Patchouli+alcohol+cancer&sort=date

Response: Thank you for your suggestion to include additional references discussing Patchouli Alcohol (PA), particularly in combination with other anticancer drugs. We appreciate your recommendation to explore PubMed as a resource. In response, we have reviewed relevant studies from the suggested link and identified several recent publications that discuss the anticancer effects of PA and its potential synergistic effects when used in combination with other therapeutic agents. We have incorporated these references into both the introduction and discussion sections of the manuscript to provide a more comprehensive context for PA’s role in cancer therapy.

We have made every effort to improve the manuscript and have implemented minor revisions. These changes do not alter the overall content or structure of the paper. Although we have not provided an itemized list of revisions, we have highlighted the changes in red in the revised manuscript for your convenience. We sincerely appreciate the time and effort that the editors and reviewers have dedicated to our work. We hope that the modifications will meet your approval. Should any further revisions be needed, we are more than willing to make the necessary adjustments. PLOS ONE is a journal of great renown, and we are hopeful that with these improvements, our manuscript will meet the high standards of your journal and be considered for publication. Once again, thank you very much for your valuable comments and suggestions.

---

## [Decision Letter · Decision Letter 1]

25 Jul 2025

Dear Dr. Pang,

Thank you for submitting your manuscript to PLOS ONE. After careful consideration, we feel that it has merit but does not fully meet PLOS ONE’s publication criteria as it currently stands. Therefore, we invite you to submit a revised version of the manuscript that addresses the points raised during the review process.

Please address the reviewers comments on a point-by-point basis and re-submit.

We look forward to receiving your revised manuscript.

Kind regards,

Joseph Alan Bauer, Ph.D.

Academic Editor

PLOS ONE

Journal Requirements:

Additional Editor Comments (if provided):

The manuscript is very well written and the quality of the experiments is superb.

Reviewers' comments:

Reviewer's Responses to Questions

**Comments to the Author**

Reviewer #1: All comments have been addressed

2. Is the manuscript technically sound, and do the data support the conclusions?

Reviewer #1: Partly

3. Has the statistical analysis been performed appropriately and rigorously?

Reviewer #1: Yes

4. Have the authors made all data underlying the findings in their manuscript fully available?

Reviewer #1: Yes

5. Is the manuscript presented in an intelligible fashion and written in standard English?

Reviewer #1: Yes

Reviewer #1: Thank you for all the corrections. I can see that authors have optimized the title and introduction and made this manuscript much better for reading. However, there are some nonnegligible issues need to be revised as below:

1. The term “network pharmacology” has been substituted in the title, but the keywords are still referring to network pharmacology. Also, the sequence of keywords needed to be re-considered.

2. The structure formula of Patchouli alcohol (PA) is not found in this manuscript. It is better to be shown with high resolution in figure 1 or figure 3.

3. The reason that the concentrations of PA are 25, 50 100 μM is not explained in the content. Please show some data to declare the origin of these concentrations.

4. In figure 4C, the figure legend says “Cycle arrest of 143B and HOS cells detected by flow cytometry. *p < 0.05 vs. control group.” But there is no data bar or * tag in the column diagram. Please check the figure and revise this mistake.

5. This research lacks of novelty but adhere to writing conventions. To improve the quality and publishing possibility, it is necessary to address more about the creativity and significance of this research.

**Do you want your identity to be public for this peer review?** For information about this choice, including consent withdrawal, please see our Privacy Policy

Reviewer #1: No

---

## [Author Response · Author response to Decision Letter 2]

22 Aug 2025

Thank you for your letter and for the reviewers’ insightful comments on our manuscript entitled “Exploration of the Anti-osteosarcoma Activity of Patchouli Alcohol Based on Network Pharmacology and In Vitro Experiments” (ID: PONE-D-24-41403). We sincerely appreciate the reviewers’ valuable feedback, which has greatly helped us to improve both the quality and clarity of the manuscript. In response to reviewers’ suggestions, we have made minor revisions, with the updated sections highlighted in red for ease of reference. We are grateful for your time and consideration, and we look forward to your further feedback on our revised submission.

Respond to the reviewer's comments:

Question 1. The term “network pharmacology” has been substituted in the title, but the keywords are still referring to network pharmacology. Also, the sequence of keywords needed to be re-considered.

Response: We thank the reviewer for this helpful observation. To align with the revised title and the study’s primary focus, we have removed “network pharmacology” from the keywords. We also conducted a manuscript-wide check and replaced or removed any residual mentions of “network pharmacology” where appropriate. For clarity and indexing, the remaining keywords have been reordered.

Revisions made in the manuscript:

Previous keywords: osteosarcoma; network pharmacology; apoptosis; Patchouli alcohol; autophagy.

Revised keywords: Osteosarcoma; Patchouli alcohol; Protein network mapping; Apoptosis; Autophagy.

Question 2.The structure formula of Patchouli alcohol (PA) is not found in this manuscript. It is better to be shown with high resolution in figure 1 or figure 3.

Response: We appreciate the reviewer’s valuable suggestion. In the revised version, we have supplemented the structural formula and molecular formula of Patchouli alcohol (PA) in Figure 3, presented with high resolution to ensure clarity and readability. We believe this addition will help readers more intuitively understand the compound investigated in this study. The manuscript has been revised by updating Figure 3 to incorporate the chemical structure and molecular formula of PA, and the corresponding figure legend has been adjusted accordingly.

Question 3. The reason that the concentrations of PA are 25, 50 100 μM is not explained in the content. Please show some data to declare the origin of these concentrations.

Response: We thank the reviewer for this constructive suggestion. The concentrations of PA (25, 50, and 100 µM) were selected according to previous reports in which this range showed significant biological activity without apparent cytotoxicity, while concentrations above 100 µM exhibited toxicity. Based on these findings, we adopted 25-100 µM as a safe and effective range for evaluating the pharmacological effects of PA in our study. To clarify this rationale, we have revised the Methods section (PA treatment) to include the following statement:

“The concentrations of PA (25, 50, and 100 µM) were chosen according to previous studies indicating that this range exerts biological activity without detectable cytotoxicity, whereas concentrations above 100 µM tend to induce toxicity1, 2”

1. Kumar, K. J. S.; Vani, M. G.; Chinnasamy, M.; Lin, W.-T.; Wang, S.-Y., Patchouli Alcohol: A Potent Tyrosinase Inhibitor Derived from Patchouli Essential Oil with Potential in the Development of a Skin-Lightening Agent. Cosmetics 2024, 11 (2), 38.

2. Lee, H. S.; Lee, J.; Smolensky, D.; Lee, S. H., Potential benefits of patchouli alcohol in prevention of human diseases: A mechanistic review. International immunopharmacology 2020, 89 (Pt A), 107056.

Question 4. In figure 4C, the figure legend says “Cycle arrest of 143B and HOS cells detected by flow cytometry. *p < 0.05 vs. control group.” But there is no data bar or * tag in the column diagram. Please check the figure and revise this mistake.

Response: We thank the reviewer for pointing this out. The raw data were correct; however, the original statistical presentation was not sufficiently clear. To improve clarity, we re-plotted the data for Figure 4C using a more intuitive format and added the appropriate statistical annotations. The revised figure now accurately reflects the data and is consistent with the figure legend.

Question 5. This research lacks of novelty but adhere to writing conventions. To improve the quality and publishing possibility, it is necessary to address more about the creativity and significance of this research.

Response: We thank the reviewer for this important suggestion. In the revised manuscript, we have strengthened the Discussion to more clearly highlight the novelty and significance of this work. Specifically, we emphasize that this is the first study to investigate patchouli alcohol in osteosarcoma, where it induces both apoptosis and autophagy through suppression of PI3K/Akt signaling. By integrating Protein network mapping, molecular docking, and experimental validation, our study provides a disease-specific mechanistic framework and reveals actionable targets that may inform future therapeutic combinations. These revisions better underscore the creativity and significance of our findings.

We sincerely appreciate the time and effort that the editors and reviewers have dedicated to our work. We hope that the revisions will meet with your approval, and we remain willing to make any further adjustments if required. PLOS ONE is a journal of high repute, and we trust that with these improvements, our manuscript will meet its rigorous standards and be considered for publication. Once again, we thank you for your valuable comments and suggestions.

---

## [Decision Letter · Decision Letter 2]

28 Aug 2025

Investigating the Anti-osteosarcoma effects of Patchouli Alcohol through protein network mapping and in vitro experiments

PONE-D-24-41403R2

Dear Dr. Pang,

We’re pleased to inform you that your manuscript has been judged scientifically suitable for publication and will be formally accepted for publication once it meets all outstanding technical requirements.

Kind regards,

Joseph Alan Bauer, Ph.D.

Academic Editor

PLOS ONE

Additional Editor Comments (optional):

Reviewer #1:

Reviewers' comments:

Reviewer's Responses to Questions

**Comments to the Author**

Reviewer #1: All comments have been addressed

2. Is the manuscript technically sound, and do the data support the conclusions?

Reviewer #1: Yes

3. Has the statistical analysis been performed appropriately and rigorously?

Reviewer #1: Yes

4. Have the authors made all data underlying the findings in their manuscript fully available?

Reviewer #1: Yes

5. Is the manuscript presented in an intelligible fashion and written in standard English?

Reviewer #1: Yes

Reviewer #1: The authors have addressed the novelty in the new manuscript as "Although PA has been studied in other cancers, its effects in OS have not previously been reported. Here, we demonstrate for the first time that PA induces both apoptosis and autophagy in OS cells while suppressing PI3K/Akt activity. By integrating network-based target prediction with experimental validation, we identified key nodes such as EGFR, HSP90AA1, and SRC, which are established contributors to OS progression and represent druggable targets. These findings expand the pharmacological profile of PA into a new disease context and provide a mechanism-based rationale for potential therapeutic combinations, including with platinum agents or pathway inhibitors. In conclusion, by uncovering multiple mechanisms, including apoptosis, autophagy, and PI3K/Akt pathway inhibition, this study expands the pharmacological profile of PA into osteosarcoma for the first time, underscoring its novelty and clinical relevance."

Thanks for all the effort made by authors for the corrections. I believe this version is better and more suitable fro publishing on PLOS ONE.

**Do you want your identity to be public for this peer review?** For information about this choice, including consent withdrawal, please see our Privacy Policy

Reviewer #1: No

---

## [Editor Report · Acceptance letter]

PONE-D-24-41403R2

PLOS ONE

Dear Dr. Pang,

I'm pleased to inform you that your manuscript has been deemed suitable for publication in PLOS ONE. Congratulations! Your manuscript is now being handed over to our production team.

Kind regards,

on behalf of

Dr. Joseph Alan Bauer

Academic Editor

PLOS ONE